# A computational pipeline to track chromatophores and analyze their dynamics

**Johann Ukrow†, Mathieu DM Renard†, Mahyar Moghimi, Gilles Laurent***

Max Planck Institute for Brain Research, Frankfurt, Germany

## eLife Assessment

The open-source software Chromas tracks and analyses cephalopod chromatophore dynamics. The software features a user-friendly interface alongside detailed instructions for its application, with **compelling** exemplary applications to two widely divergent cephalopod species, a squid and a cuttlefish, over time periods large enough to encompass new chromatophore development among existing ones. It demonstrates accurate tracking of the position and identity of each chromatophore. The software and methods outlined therein will become an **important** tool for scientists tracking dynamic signaling in animals.

**\*For correspondence:**
g.laurent@brain.mpg.de

†These authors contributed equally to this work

**Competing interest:** The authors declare that no competing interests exist.

**Abstract** Cephalopod chromatophores are small dermal neuromuscular organs, each consisting of a pigment-containing cell and 10–20 surrounding radial muscles. Their expansions and contractions, controlled and coordinated by the brain, are used to modify the animal's appearance during camouflaging and signaling. Building up on tools developed by this lab, we propose a flexible computational pipeline to track and analyze chromatophore dynamics from high-resolution videos of behaving cephalopods. This suite of functions, which we call CHROMAS, segments and classifies individual chromatophores, compensates for animal movements and skin deformations, thus enabling precise and parallel measurements of chromatophore dynamics and long-term tracking over development. A high-resolution tool for the analysis of chromatophore deformations during behavior reveals details of their motor control and thus, their likely innervation. When applied to many chromatophores simultaneously and combined with statistical and clustering tools, this analysis reveals the complex and distributed nature of the chromatophore motor units. We apply CHROMAS to the skins of the bobtail squid *Euprymna berryi* and the European cuttlefish *Sepia officinalis*, illustrating its performance with species with widely different chromatophore densities and patterning behaviors. More generally, CHROMAS offers many flexible and easily reconfigured tools to quantify the dynamics of pixelated biological patterns.

## Introduction

Animal skin color can change adaptively thanks to the migration or direct control of specialized pigment-containing cells. In ectothermic animals, these cells are called chromatophores. They often each contain one of several pigments, enabling elaborate patterns, colors (as in fish or reptiles), and in some cases, such as cephalopods, communication and camouflage. Coleoid cephalopods evolved the unique ability to change in an instant the patterning, color, and 3D texture of their skin through the neural control of their chromatophores. Cephalopod chromatophores are elaborate neuromuscular organs composed of a cell containing an elastic sacculus filled with pigment granules, surrounded by a set of radial muscles, themselves controlled by motor neurons located in the brain (*Cloney*

*and Florey, 1968*; *Messenger, 2001*). The skin of some species of cuttlefish and octopus possesses millions of such chromatophores at a density of 230 per mm$^2$ in adults. As biological analogs of pixels on a screen, chromatophores operate by varying their size (and thus effective reflectance) rather than photon emission. Chromatophores, together with other specialized cells such as iridophores and leucophores, enable these cephalopods to display a variety of patterns and colors that play crucial roles for camouflage and intra- and inter-specific signaling (*Hanlon et al., 1988*).

The cephalopod camouflage system operates as a perception-processing-projection pipeline, where visual input from the environment is analyzed by the nervous system and reproduced as a statistical approximation on its skin. For this reason, these skin displays offer a quantifiable readout of the animal's perceptual state and have fascinated scientists for decades. However, studying chromatophore activity over chromatophore populations is challenging because of their small size (ranging from 5 to 100 μm when retracted, depending on the species), their vast numbers (thousands to millions) over large surfaces (hundreds of cm$^2$), the inherent flexibility and motion of cephalopod skin (due to respiration, for example), and the absence of tools to track their state in real time. While researchers did, in a few studies, quantify skin patterns from still images of the animals, chromatophore resolution and tracking of patterns over time were not achieved (*Barbosa et al., 2008*; *Hanlon et al., 2009*; *Shohet et al., 2007*). Physiologists, however, did study individual or a few chromatophores in action, but this approach rested heavily on non-automated measurements, with limited precision and scalability (*Packard, 1982*), and did not address the issue of camouflage directly.

This laboratory recently developed automated and quantitative approaches to circumvent these limitations to study cephalopod camouflage and related behaviors (*Reiter et al., 2018*; *Woo et al., 2023*). The tools developed there have been rewritten and expanded in an easy-to-use software suite, which we call CHROMAS, the object of the present paper. CHROMAS is a high-resolution analysis software that aims to provide an objective quantification of chromatophore-population behavior over periods of time as long as months, using video recordings of live cephalopod skin as an input. While CHROMAS retains the foundational concepts introduced in *Reiter et al., 2018* and *Woo et al., 2023*, the tools have been rewritten to improve performance and usability. New functionalities have been added to extend the temporal range and spatial resolution of the analysis. In particular, we introduce the semi-automated tracking of chromatophore identity over development and the segmentation of individual chromatophores by analysis of deformation anisotropy. Although developed to track and analyze cephalopod skin patterning, these tools can be used with the skin of other species, and more generally, with any comparable time series of images.

CHROMAS reads video files and extracts chromatophore properties based on known biological features, derived from prior experimental work.

## Identity and position

The relative spatial arrangement of chromatophores is anatomically fixed in the skin (*Packard, 1982*) because they are anchored to a stable and appropriately stiff extracellular matrix. Their neighboring chromatophores, however, change as the animal grows, because new chromatophores are continuously added between older ones and change color as they age (*Reiter et al., 2018*). CHROMAS keeps track of chromatophore identity over time despite these changes.

## Expansion state

When radial muscles (attached distally to the extracellular matrix and proximally to a chromatophore) contract, the chromatophore's cytoplasmic membrane and internal pigment sack expand; inversely, when those muscles relax, the pigment sack retracts passively due to its elasticity. Because the chromatophore muscles are controlled by motor neurons (*Florey and Kriebel, 1969*), chromatophore size variations offer an indirect but objective and quantifiable readout of neural activity (*Reiter et al., 2018*) at video sampling rate.

## Anisotropy of expansion and detailed innervation

The radial muscles controlling a chromatophore can collectively be innervated by more than one motor neuron (*Florey, 1969*). Chromatophore expansion is then determined by a set of potentially independent forces, which can result in irregular or anisotropic chromatophore deformations. Analysis

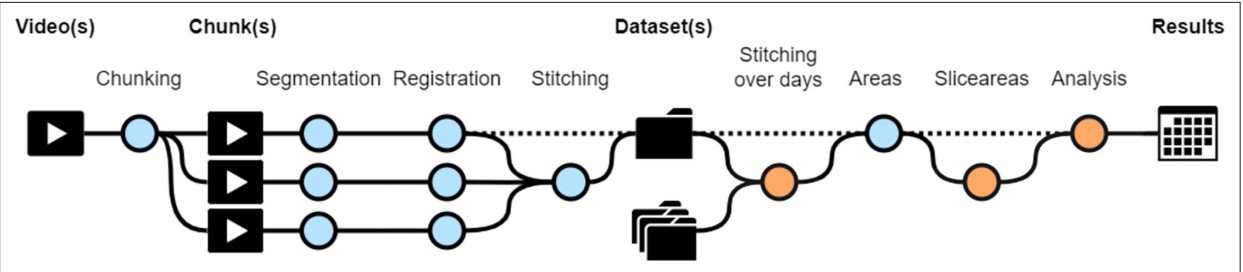

**Figure 1.** Workflow diagram of CHROMAS. Dotted lines indicate optional steps in the process, allowing flexibility depending on experimental requirements. Orange dots represent new functions that did not exist in the initial versions of the pipeline (*Reiter et al., 2018*; *Woo et al., 2023*). Each step of the pipeline can be run independently, providing users with the flexibility to execute specific stages as needed. Additionally, CHROMAS includes options to generate visual output videos, enabling users to assess and validate the accuracy and quality of the result at every step.

of this anisotropy can thus inform us of the fine innervation of individual chromatophores. CHROMAS provides this information.

We hope that these tools will become valuable resources to address complex biological questions regarding chromatophore neural control, development, biomechanics, and more generally, cephalopod perception and fine motor behavior. More broadly, these tools should be useful also for detailed moving-image analysis in other systems.

## Results

CHROMAS provides a versatile framework to investigate a wide range of biological questions. Each tool can be used independently of the others to solve specific tasks, while their combined use enables

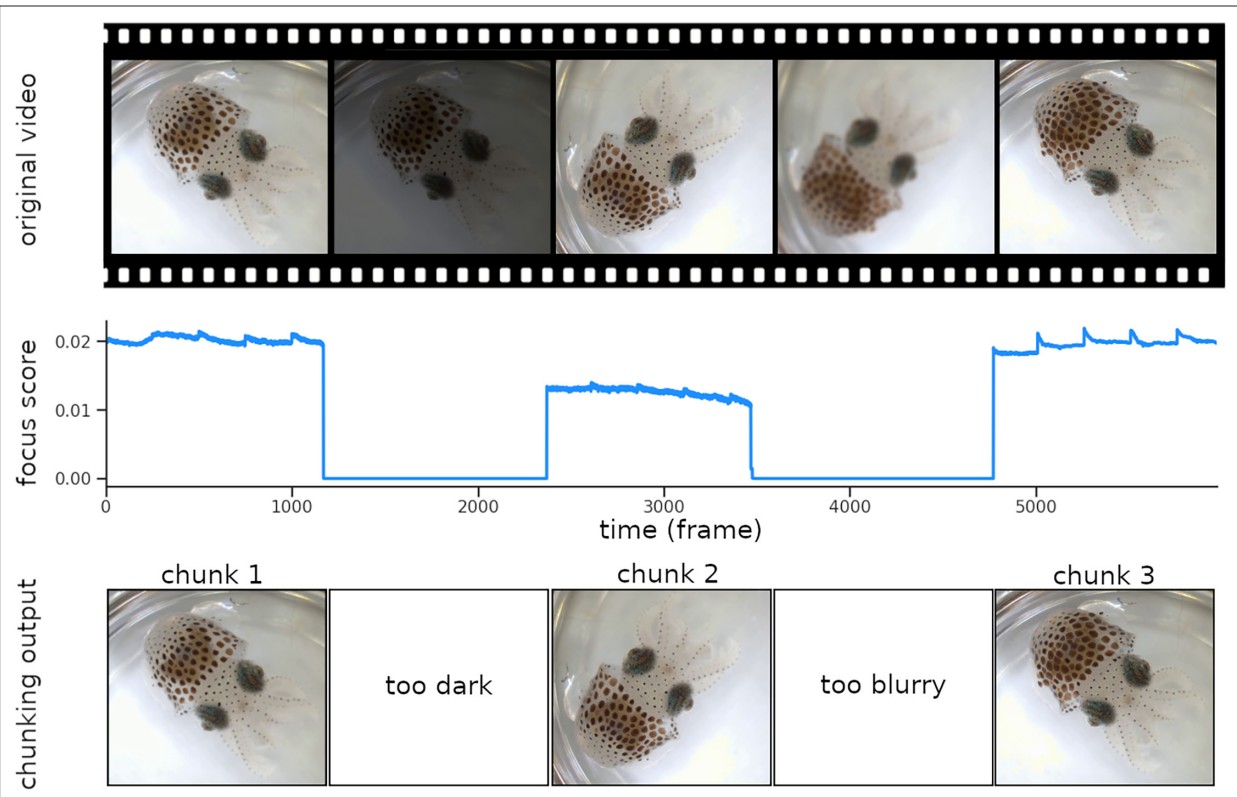

**Figure 2.** Chunking process. The video frames are sorted based on their focus score. Images that are too dark or too blurry are removed from the dataset, and chunks made of consecutive sharp frames are created. In the figure above, the score decreases due to a brightness drop at frame 1200 and to the subject going out of focus around frame 3500, resulting in three distinct chunks. Continuity between chunks is reestablished during the 'Stitching' process (see *Figure 5*).

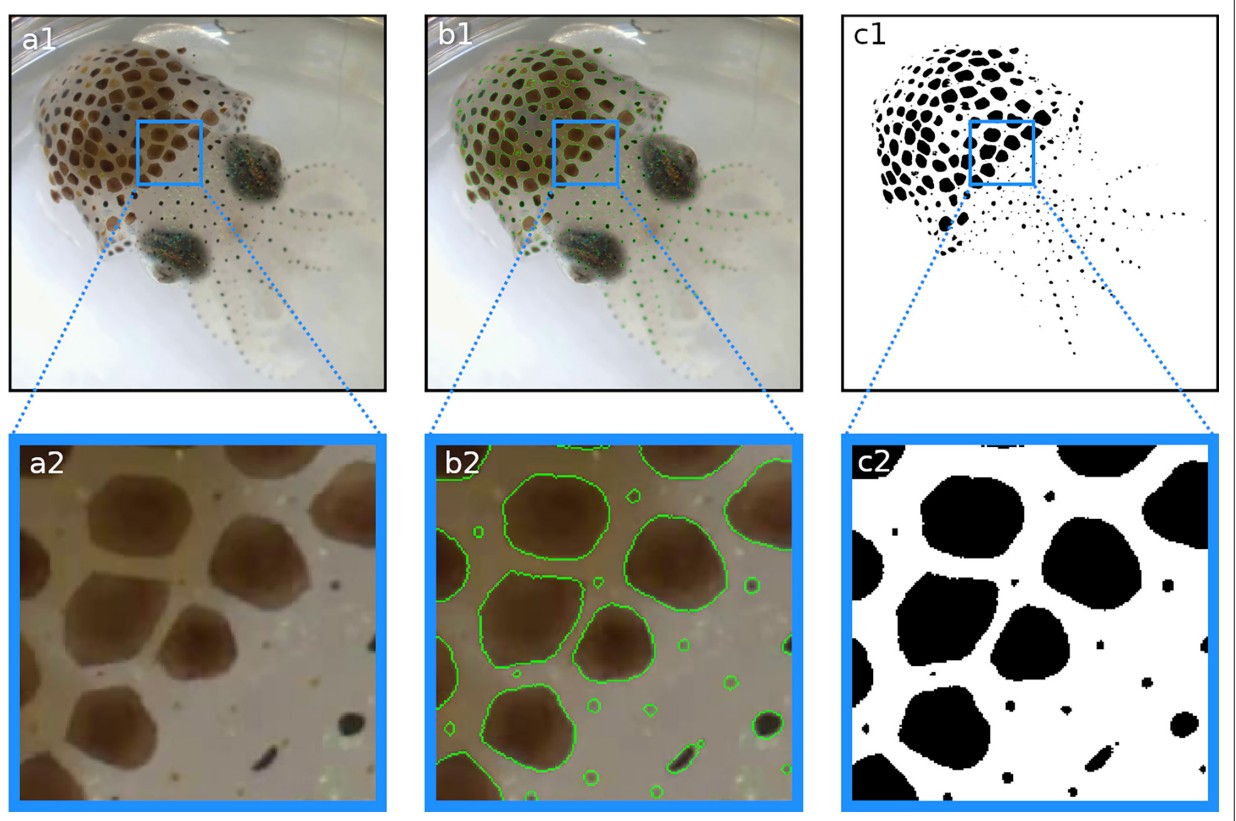

**Figure 3.** Segmentation results. In each frame, each pixel is classified as either part of a chromatophore or not. (**a1, a2**) Original frame before segmentation. (**b1, b2**) Original frames with edges of segmented chromatophores in green. (**c1, c2**) Binary segmentation with chromatophores in black and background in white. Note that even the smallest chromatophores, though faint and difficult to discern on the original footage, are segmented.

the construction of elaborate workflows. The subsequent sections provide examples of specific tasks that, when combined sequentially, constitute a powerful workflow (see *Figure 1*), as demonstrated in *Reiter et al., 2018* and *Woo et al., 2023* with *Sepia officinalis*. *Figures 1–8* illustrate the main functions with *Euprymna berryi*, because of its small size and limited number of chromatophores. All functions have been developed to work with larger species with greater chromatophore numbers and density, such as *S. officinalis*, as shown later in *Figure 9*.

We provide a comprehensive tutorial guide along with sample video files to test the pipeline. Links to these resources are available in the 'Data availability' section. The entire pipeline can be run with the command below, and individual task command lines are included as each step of the pipeline is detailed.

Command line:
> chromas run /path/to/example.mp4

## Chunking videos into usable segments

Analyzing long video recordings of behaving-cephalopod skin can be frustrating because of occasional movement blur, de-focusing, and obstruction. To address this, we divide the video into continuous segments which we call 'chunks', consisting of consecutive frames where the animal's mantle, or a part of it, is both visible and in focus. An animal, or a field, is considered to be in focus if the edges of its chromatophores are clearly defined. The identification of these chunks can be accomplished using focus statistics, which are numerical measures indicative of the sharpness of an image (*Figure 2*, focus score). A method often used is the difference of Gaussians, which involves subtracting a blurred version of an image (produced by convolving the image with a Gaussian kernel with a large standard deviation) from a less blurred version of the original image (using a smaller standard deviation). The resulting image emphasizes areas of steep intensity change, such as edges, while suppressing

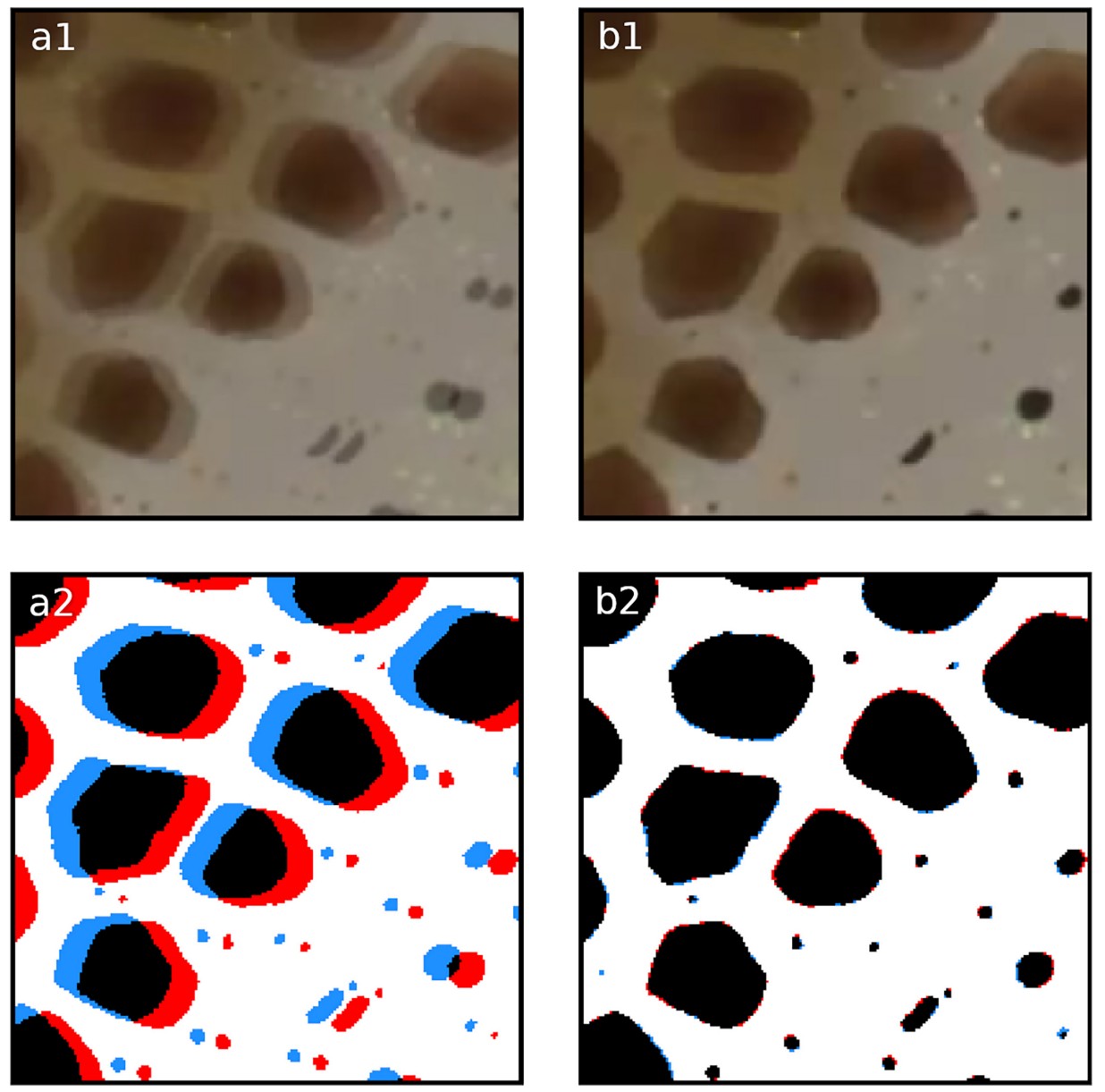

**Figure 4.** Registration. Overlay of two frames 20 frames apart within the same chunk. Panels (**a1, a2**) illustrate the misalignment of two chromatophores (red and blue) before registration. Panels (**b1, b2**) show the same frames after automated registration. The registration process compensates for small movements or drifts over time, ensuring correct identification and tracking of the chromatophores over time.

low-frequency variations, making it well suited for detecting the sharp boundaries of chromatophores. The standard deviations of the two Gaussian kernels were chosen such that their difference matched the typical size of chromatophores, to increase the visibility of their edges. The input video is then cut into shorter clips—the chunks (*Figure 2*, chunking output). This step can be adapted to exclude other types of unwanted frames based on other parameters, such as low brightness, the absence of a fluorescent tag in the frame, or motion blur.

Command line:
> chromas chunk /path/to/example.mp4

## Chromatophore segmentation and color classification

It is essential to identify and isolate individual chromatophores accurately in video frames before a detailed analysis can begin. 'Segmentation' refers to the process of generating a binarized image,

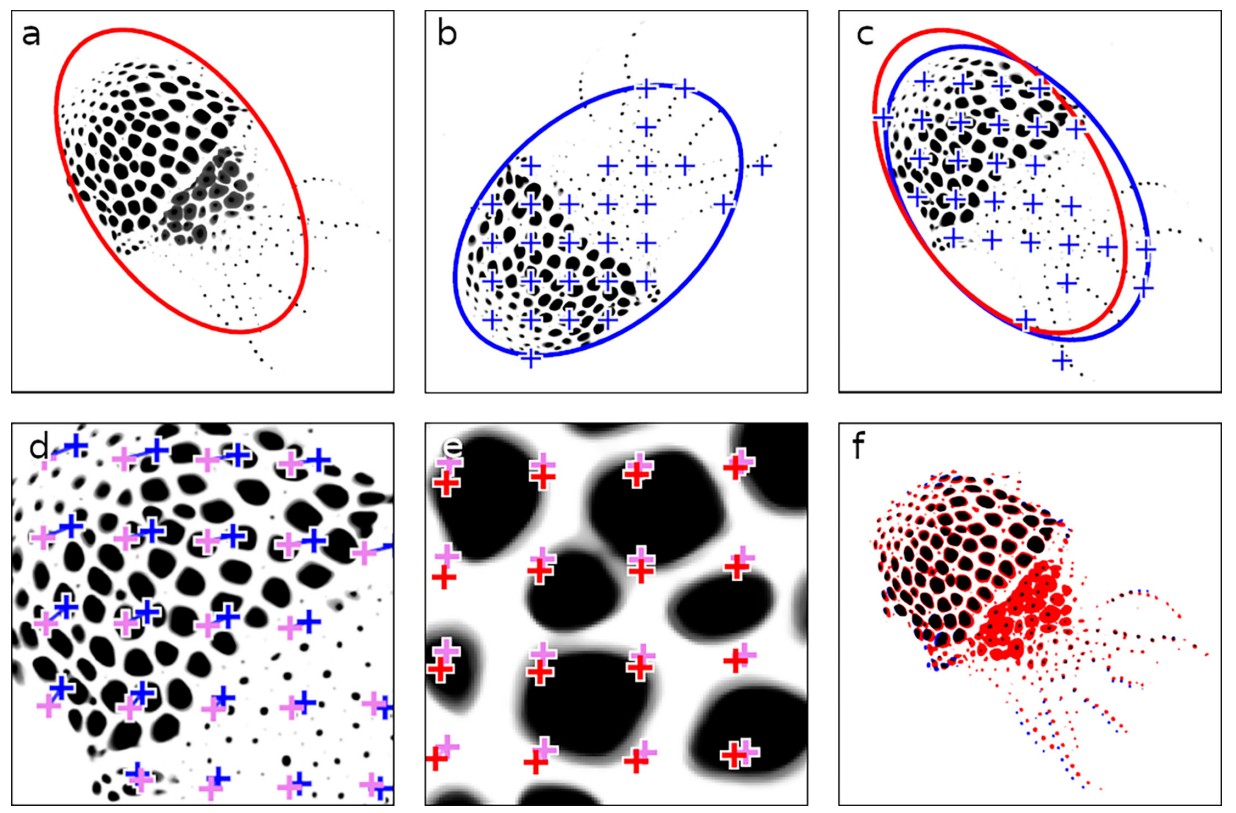

**Figure 5.** Stitching video chunks. This figure illustrates the stepwise refinement process for stitching adjacent video chunks, ensuring precise alignment critical for downstream analyses. (**a**) Masterframe of chunk 1, with an initial ellipse fit shown in red. (**b**) Masterframe of chunk 2, featuring the ellipse fit and a grid of points for fine alignment in blue. (**c**) Chunk 2 aligned on chunk 1 using the ellipse-fitting method. (**d**) First round of fine alignment, maximizing phase correlation between patches surrounding the points. Points prior to fine alignment are shown in blue, and their adjusted positions post-alignment are in pink. (**e**) Second round of fine alignment, from pink to red. (**f**) Final overlay between chunk 1 and the fully aligned chunk 2. In black are the chromatophore areas common to both chunks; in red and blue are chromatophore areas that are unique to chunks 1 and 2, respectively.

where each pixel is classified as either part of a chromatophore or part of the background (*Figure 3*). Chromatophore classification by color is also supported. Segmentation relies on deep learning models trained on manually annotated data. We provide two trained models: one for binary (yes, no) and one for chromatophore-color segmentation (dark, orange, yellow), trained on annotated data of *S. officinalis* and *E. berryi* under different lighting conditions and camera setups. Furthermore, these models can be fine-tuned on custom data, and various models can be trained from scratch on custom data to enable research on different species. Available models include Fully Convolutional Networks (*Long et al., 2015*), DeepLabV3 (*Chen et al., 2017*), or U-Net (*Ronneberger et al., 2015*), with ResNet50, ResNet101 (*He et al., 2016*), or MobileNetv3-Large (*Howard et al., 2019*) serving as backbones. Alternatively, segmentation can also be achieved by a random forest classifier or a combination of both, followed by 'majority vote'.

Command line:
> chromas segment /path/to/example.dataset

## Compensating for movement: Registration

To measure a chromatophore's activity (deformations), its identity must be correctly transferred across frames; this is a difficult task if its coordinates change from frame to frame. Chunk selection ensures that video segments with blur due to rapid motion have been excluded from the data. However, chromatophores often change position from one frame to another due to slower or more subtle movements such as breathing, slow drifting, or skin deformation, while remaining in focus. Therefore, what we call the 'Registration' operation compensates for the animal's movements to maintain a consistent location of individual chromatophores over time. For this, the Lucas–Kanade optical flow algorithm

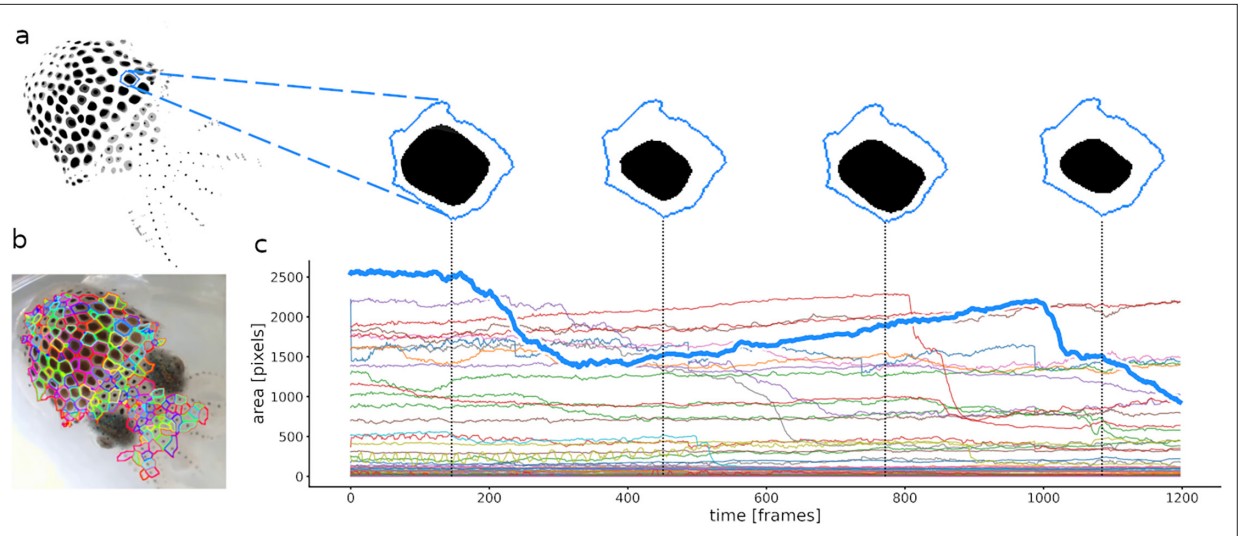

**Figure 6.** Calculation of chromatophore areas. (**a**) Masterframe and zoom on the segmentation of one chromatophore's contractions and expansions. (**b**) Cleanqueen. Each color represents a different chromatophore territory. (**c**) Surface area of each segmented chromatophore over time (colors as in b). The chromatophore in a is highlighted in the thick blue line. This approach measures the total area occupied by a chromatophore within its defined territory and offers no information about its shape.

(**Lucas and Kanade, 1981**) tracks points that are initially randomly sampled on the first frame, from frame to frame. Full displacement maps are then interpolated from the displacement of these tracking points using a moving-least-squares algorithm. This results in all the frames being registered with the first frame of the video as reference (**Figure 4**). To ensure accurate registration, tracking points that move implausibly far between frames are automatically discarded, and the registration process is halted if too many points are lost, preventing poor-quality mappings from being used in the analysis pipeline.

Command line:
> chromas register/path/to/example.dataset

## Stitching across video chunks

To create a cohesive dataset, the correct identity of each chromatophore must be tracked across chunks. The absence of visual information between chunks (by definition of the chunks), however, renders standard optical flow techniques unsuitable for this task. A new technique was developed for this end involving the following steps. All registered frames within a chunk are averaged to produce a descriptive image of the position and color of each chromatophore. This average is referred to as the 'masterframe' of that chunk (**Figure 5a**, without the ellipse). If the animal's body is fully visible, an initial alignment of the masterframes of successive chunks is achieved through automatic ellipse fitting around the animal's silhouette (**Figure 5a–c**). Otherwise, alignment is performed after the user manually selects matching points between masterframes. Subsequently, displacement vectors are calculated for a regular grid of points sampled over the animal's body by maximizing phase correlation between image patches surrounding these points (**Figure 5b–d**). Full displacement maps are then interpolated using a moving-least-squares algorithm. The accuracy of these mappings is quantified using the reprojection error metric, as detailed in the methods section, and chunks with high reprojection error are excluded from further analysis.

Command line:
> chromas stitch /path/to/example.dataset

## Tracking chromatophore size

Now that the chromatophores have been aligned across frames and chunks, their identity can be tracked and their expansion quantified (**Figure 6a**). First, a 2D array, termed 'Cleanqueen', is generated, that delineates individual chromatophore territories, that is, the space that each chromatophore can occupy (**Figure 6b**). Using inverse registration maps, the cleanqueen is mapped onto the

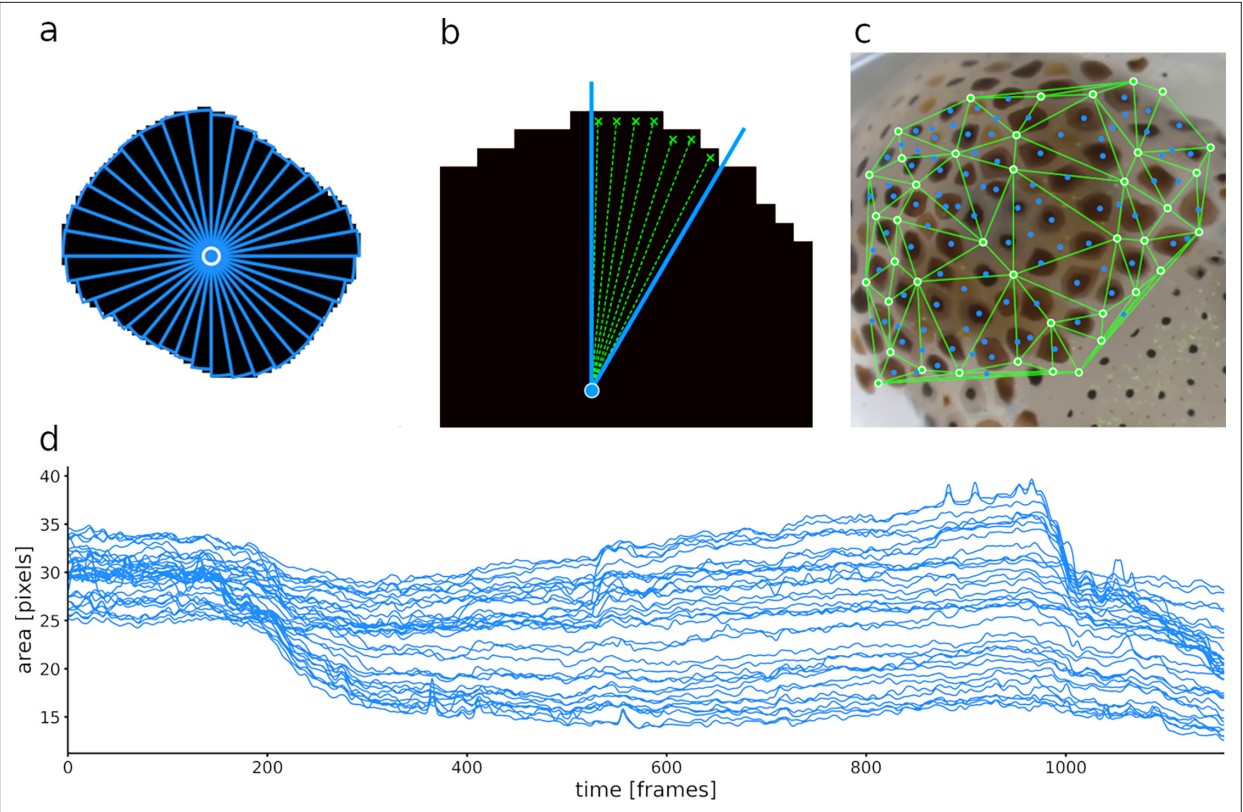

**Figure 7.** Calculation of slice surface areas. (**a**) A single chromatophore (the same one as in *Figure 6*) is divided radially into 36 slices. (**b**) Close-up on one of the 36 slices, indicating how its surface area is estimated. (**c**) Chromatophore 'epicenters' (see text) in blue and motion markers (see text) in green, with edges for triangulation. (**d**) Plot representing the surface areas of the 36 slices of this chromatophore over time. Note that the dynamics of a chromatophore are now described by 36 values per frame, rather than one, as in *Figure 6c*. Note also that the top half of the traces describing this chromatophore deviates clearly from the others around frames 550 and 960, indicating their differential control. This fine-grain description of chromatophore deformations over time is used to reveal the fine details of their individual and collective motor control.

unregistered segmentation frames and the identity of each chromatophore can be tracked. The area of each chromatophore is then calculated as the number of pixels forming the largest connected component inside the chromatophore's corresponding territory (*Figure 6c*).

Command line:
> chromas area /path/to/example.dataset

## Tracking anisotropic chromatophore activity

For most uses, analysis of the global chromatophore expansion is sufficient. There exist conditions, however, where a high-resolution analysis of chromatophore deformations is useful. We will show that it can reveal the multiple innervation of individual chromatophores and the fine and distributed nature of motor units.

Quantifying the kinematics of an ever-changing shape on a surface which is also subject to deformations is not a trivial task. Whereas chromatophores can be viewed as monochromatic objects that change shape over time, they reside on a surface (the skin) that itself undergoes both local and global deformations, such as mantle muscle contractions or movements caused by breathing, locomotion, or external forces such as water currents. Therefore, an accurate description of the kinematics of a chromatophore can be done only once it has been disentangled from the distortions of the skin. We thus addressed the need to accurately stabilize (i.e., compensate for the movements and deformations of) the background. Registration maps were not suitable for this task, because an optic flow approach would try to correct for chromatophore expansion. We thus developed a different type of registration. This step takes advantage of the fact that the 2D configuration of chromatophores in their most contracted state tends to be fixed in space. The center of mass of chromatophores that remain small

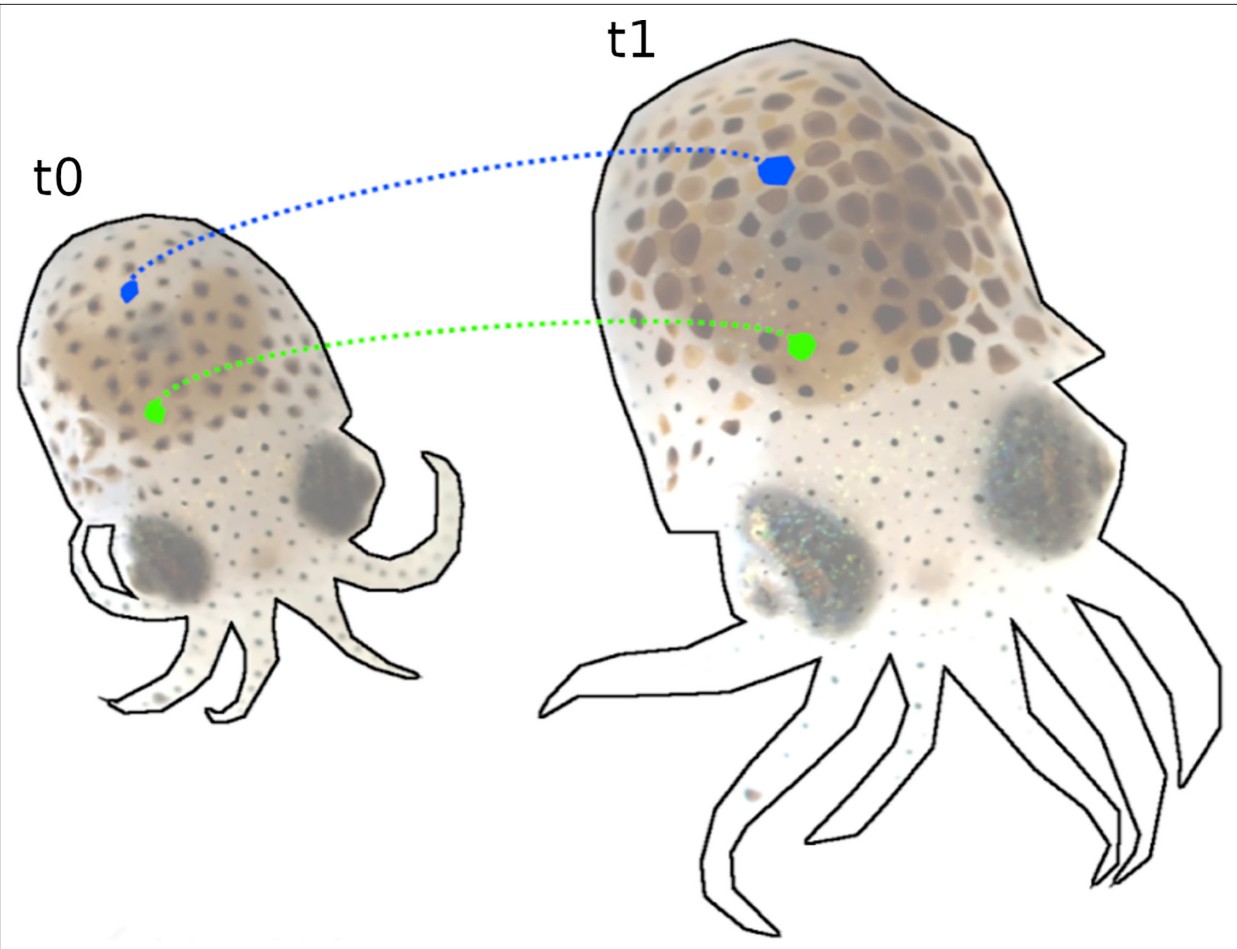

**Figure 8.** Tracking of chromatophores over 12 days of development ($t_0$ = 7 days post hatching (dph) and $t_1$ = 21 dph). During this time, the number of chromatophores nearly doubled, with many changing chromatic properties. Despite these changes and intermittent filming, our pipeline reliably tracks individual chromatophores, as illustrated with two arbitrarily selected chromatophores, in blue and green.

and constant in size throughout the video (typically, chromatophores at early or intermediate stages of development, *Reiter et al., 2018*) can thus be viewed as inert points on the animal's skin, and their displacement over frames can then be used to derive skin deformation. We call these points 'motion markers'. Once this operation has been accomplished, the details of a chromatophore's deformations can be obtained.

To quantify these deformations, each chromatophore is divided into radial slices, and the area within each slice is calculated independently (*Figure 7a*). To slice a chromatophore radially, its 'epicenter' (the starting point of its expansion, blue dots in *Figure 7a–c*), must be identified and tracked over frames. This point cannot be the center of mass of the chromatophore, because anisotropic deformations would wrongly shift this point between frames. Instead, a chromatophore's epicenter is computed following the logic of the previous paragraph: it is calculated as the center of mass at the frame where the chromatophore is in its most contracted state. Because the spatial relationships between epicenters are constant in time, their coordinates can be stored relative to the coordinates of their respective motion markers (*Figure 7c*, green dots), that is the three nearest motion markers spanning a triangle around the chromatophore in a Delaunay triangulation (*Figure 7c*, green triangles represent the Delaunay triangulation). Relative coordinates are stored as barycentric coordinates $(\lambda_1, \lambda_2, \lambda_3) \in R^3$, such that the absolute coordinates $c$ of the epicenters can be calculated as $c = \lambda_1 \cdot a_1 + \lambda_2 \cdot a_2 + \lambda_3 \cdot a_3$ from the absolute coordinates $a_1, a_2, a_3$ of their motion markers. The combination of these local coordinate systems, along with a sufficiently dense distribution of motion markers, enables us to eliminate the global and local deformations of the skin and to successfully track the positions of all epicenters.

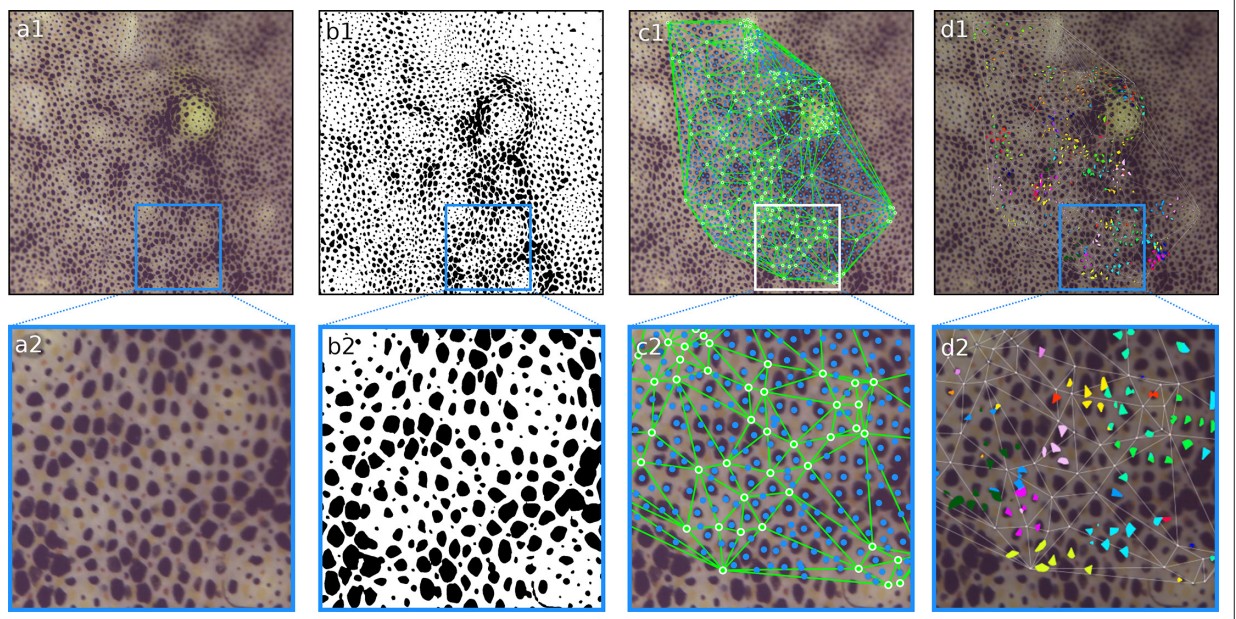

**Figure 9.** Example of 4K footage of *Sepia officinalis* processed through the CHROMAS pipeline. (**a1, a2**) Original video frame. (**b1, b2**) Binary segmentation with chromatophores in black and background in white. (**c1, c2**) Epicenters of chromatophores in blue and motion markers with triangulation in green. (**d1, d2**) Putative motor units calculated using the Affinity Propagation algorithm. In this example, Principal Component Analysis–Independent Component Analysis (PCA–ICA) revealed independent components influencing the activity of each chromatophore. We found 4.68 ± 1.41 (mean ± std) ICs per chromatophore (n = 1616). Those independent components were then clustered together based on their covariation and plotted with the same color. For visualization purposes, only clusters of size 8 and 9 are plotted. In this segment, we found 10.37 ± 14.23 (mean ± std) chromatophores per cluster, with 90% of clusters consisting of less than 20 chromatophores. This example highlights CHROMAS's ability to process complex datasets, such as footage of *Sepia officinalis*, which exhibits a high density of chromatophores and very active dynamics correlated with its camouflage and other behaviors.

In addition, the motion markers are used to keep the orientation of the slices constant over frames, which is necessary to track the identity of each slice.

By default, chromatophores are sliced into 36 sectors. This number was selected based on the Nyquist–Shannon sampling theorem (*Shannon, 1948*), and on histological analysis showing that chromatophores in *E. berryi* and *S. officinalis* are each controlled by approximately 10–15 radial muscles. This number of slices struck an acceptable balance, ensuring a high enough resolution to capture realistic shape changes, while keeping computational demands manageable. To reduce measurement errors induced by pixel discretization, slice areas were computed by averaging the distance from the border pixels of the slice to the epicenter $r$ and then computing the respective area as $area = \pi \cdot r^2/n$, see *Figure 7b*.

Command line:
> chromas slice /path/to/example.dataset

## Long-term chromatophore tracking

Tracking individual chromatophores over extended periods (days or weeks) is necessary to examine, among others, functional and developmental issues of control. While standard stitching techniques (see 'Stitching') suffice for datasets captured in close temporal proximity (hours), they fail for longer time periods, due to significant changes in animal size, chromatophores color, chromatophore numbers, and occasionally the disappearance of existing ones.

To address this, CHROMAS includes an option for a manual pre-alignment step prior to automated stitching. This process is facilitated by an intuitive graphical interface (based on *matplotlib* and *tkinter*) allowing users to select corresponding points between two frames from videos captured at different stages. The interface has been developed with usability in mind, providing options such as zoom, rotation, and an 'undo' function. These selected points are then utilized to generate warping maps using either the moving-least-squares algorithm or thin plate splines (*NumPy* and *SciPy*), providing a precise

**Table 1.** Performance on a workstation with a 10-core Intel Core i9-10900X with 64 GB RAM and a NVIDIA RTX A4000 GPU.
Runtime in milliseconds per frame (mean ± s.d.).

| Resolution | Number of chromatophores (ms/frame) | Segmentation (neural net) (ms/frame) | Registration (ms/ frame) | Tracking chromatophore size (ms/frame) | Tracking anisotropic activity (ms/frame) |
|---|---|---|---|---|---|
| SD | 146 | 5.38 ± 0.1 | 18.61 ± 0.1 | 13.01 ± 1.4 | 4.98 ± 0.9 |
| HD | 566 | 22.31 ± 0.7 | 62.87 ± 2.3 | 140.44 ± 1.0 | 135.24 ± 0.9 |
| k | 2842 | 986.21 ± 10.4 | 611.74 ± 5.4 | 744.69 ± 2.5 | 495.17 ± 2.8 |

initial alignment that significantly improves the continuity and accuracy of subsequent stitching and analysis (*Figure 8*).

Command line:

> chromas superstitch /path/to/example.dataset /path/to/later_example.dataset

## Built-in analysis tools and additional features

CHROMAS also offers a series of built-in analysis tools. Clustering algorithms, for example, are particularly useful to study how motor neurons control small groups of chromatophores, or sets of sectors of chromatophores, potentially revealing patterns of co-innervation and synchronized activity across specific regions of the skin. Our software enables users to perform Principal Component Analysis (PCA), Independent Component Analysis (ICA), and clustering analysis, including, but not limited to, Affinity Propagation (AP) and Hierarchical Density-Based Spatial Clustering of Applications with Noise (HDBSCAN). PCA and ICA are instrumental in disentangling the influence of motor neurons on individual chromatophores, while clustering algorithms uncover the broader distribution of motor neuron influence across larger scales (*Figure 9*). Such tools can be found under 'Methods—Clustering motion correlation'.

Furthermore, CHROMAS integrates advanced tools for analyzing full-body patterns, enabling studies on camouflage effectiveness and pattern dynamics. The software employs the Structural Similarity Index to measure perceptual similarity between biological and background patterns, implemented using the *scikit-image* Python package. Patterns are embedded into a lower-dimensional 'pattern space' using dimensionality reduction techniques like PCA and t-Distributed Stochastic Neighbor Embedding, facilitated by the *scikit-learn* and *openTSNE* libraries. Temporal dynamics of patterns are analyzed by plotting trajectories in this space using the *pandas* and *matplotlib* packages, revealing the speed, directionality, and stability of pattern transitions, as described in *Woo et al., 2023*.

Command line:

> chromas analyse XXX /path/to/example.dataset

## Performance

Performance was benchmarked using a 5-min-long 4K video (shot at 20 frames per second, totaling 6000 frames and 494.5 MB) featuring a close-up view of live skin of *S. officinalis*. The benchmark also included two modified versions of the same video, cropped to Full-HD (2000 × 2000) and SD (1000 × 1000) resolutions. Cropped videos preserve the original pixel density per chromatophore but include fewer chromatophores for analysis.

**Table 2.** Performance on a laptop with a 4-core Intel Core i5-1135G7 with 16 GB RAM and no NVIDIA GPU.
Runtime in milliseconds per frame (mean ± s.d.).

| Resolution | Number of chromatophores (ms/frame) | Segmentation (lookup) (ms/frame) | Registration (ms/frame) | Tracking chromatophore size (ms/frame) | Tracking anisotropic activity (ms/frame) |
|---|---|---|---|---|---|
| SD | 146 | 11.55 ± 1.81 | 4.67 ± 0.63 | 14.56 ± 5.1 | 9.31 ± 0.6 |
| 4k | 2842 | 1679.62 ± 251.8 | 1490.94 ± 249.9 | 3072.94 ± 121.6 | 1488.25 ± 84.0 |

*Tables 1 and 2* document the processing time per frame (in milliseconds) for individual tasks, based on different image sizes and hardware configurations. The hardware categories include a workstation (with a 10-core Intel Core i9-10900X with 64 GB RAM and a NVIDIA RTX A4000 GPU), as well as a laptop (with a 4-core Intel Core i5-1135G7 with 16 GB RAM and no NVIDIA GPU).

While the runtime performance of all tasks (except for stitching) is influenced by hardware capabilities, video length, and resolution (in this particular order), the performance of areas and slice-areas calculation and analysis is also influenced by the number of visible chromatophores in the footage. Furthermore, the performance of stitching depends only on the hardware, the resolution, and the number of chunks.

## Discussion

We developed CHROMAS to be an easy-to-use, flexible, and expandable tool for the objective and statistical quantification of the dynamics of chromatophore activity. Most of the steps in CHROMAS are built upon methodologies established in previous work (e.g., *Reiter et al., 2018* and *Woo et al., 2023*), in particular chunking, segmentation, registration, stitching, and area calculation (*Figure 1*). These processes were entirely restructured and rewritten in this study to prioritize efficiency, modularity, and ease of use. Beyond these changes and improvements, CHROMAS introduces novel features that further refine the scope of achievable chromatophore analysis using computer vision methods.

First, our results achieve high levels of spatial resolution. Indeed, our anisotropy analysis captures the dynamics of the smallest cellular components of the chromatophore organs—their radial muscles. The numbers of motor neurons innervating single chromatophores and the numbers of chromatophores innervated by a single motor neuron, both estimated via our image analysis methods, match those measured earlier by direct and minimal electrical stimulation of terminal chromatophore nerves (*Reiter et al., 2018*), providing confidence that our new results are accurate. This image-based approach should thus enable one to describe the precise innervation patterns of very large fields of chromatophores, and their evolution in time, at chromatophore-motor-neuron resolution. This level of description for a complex motor output and behavior, especially with tools that require no external indicator is, to our knowledge, unique.

Second, our semi-automatic methods for tracking chromatophore identity overcome key challenges in long-term monitoring, the biggest one of which is the continuous addition of new chromatophores as the animal develops. This development enables the tracking of chromatophores over very long times (weeks to months), thus spanning the entire lifespan of an animal.

### Scalability

Tasks within CHROMAS are designed for efficient parallelization. This enables scaling to the full capacity of single machines as well as execution on high-performance supercomputers. In addition, 'larger-than-memory' execution is supported, allowing for the processing of datasets whose size exceeds the available RAM (e.g., with 8K-resolution images). CHROMAS uses an approach to parallelization that does not depend on the hardware, ensuring code maintainability and versatility. This design allows the same script to run efficiently on various hardware configurations, from single-core machines to multi-core systems and clusters. Tasks are parallelized along multiple axes (chunk-based, temporal, and spatial). While temporal parallelization is mainly used, some tasks (segmentation, registration, areas, slice areas) are parallelized along the spatial dimensions as well. For cluster-level parallelization, initial setup on the cluster is required. CHROMAS supports most high-performance computing job schedulers found in scientific research environments, such as SLURM, SGE, TORQUE, LSF, DRMAA, and PBS. Hence, the spatial scale and temporal resolution of our analyses are bound less by the capabilities of CHROMAS than by the quality of the image data and the available computational power.

### Usability

CHROMAS can operate in four distinct modes, adapted to user needs. The software can be controlled by a graphical user interface (GUI), enabling quick and easy operation without programming knowledge and widening the pool of potential users. It can also be controlled by a command line interface, offering additional flexibility as well as automation and scripting capabilities. We provide an extensive low-level application programming interface, enabling custom workflows, integration with external

tools, and the development of extensions. Last, we provide Jupyter notebooks facilitating interactive data analysis for common use cases.

The software can be installed either via the PyPI package manager, through the executable installer (.exe), or manually, using the Git repository. Extensive documentation is supplied and hosted.

Both high-level parameters, like algorithmic choices or storage configurations, and lower-level parameters, like concrete parameters of algorithms, are configured for ease of use within a single YAML file, which can also be modified through the GUI.

## Known limitations

CHROMAS analyzes the activity of chromatophores by dividing the mantle into small mosaics (in the 'cleanqueen'), where each chromatophore is assigned its own territory. Because these territories are discrete—they belong to only one chromatophore at a time—there can be species, ages, and conditions in which the chromatophores expand so much that they encroach on each other's space and overlap partly. Currently, our software is not able to resolve these overlaps.

Another known limitation concerns the biological assumptions underlying the current version of CHROMAS. The pipeline is designed for surfaces that remain reasonably planar and undergo deformations primarily in two dimensions. In cephalopods such as octopuses, in which the skin can undergo substantial three-dimensional morphological changes, analyzing chromatophore dynamics may require complementary three-dimensional tracking of the skin surface to correct for out-of-plane deformations and maintain accurate measurement of chromatophore activity.

## Recommended video parameters for optimal use of CHROMAS

The performance of CHROMAS depends on the quality of the input videos. Although the pipeline analyzes each frame independently and has no frame rate requirement, we recommend recording at 20 frames per second at least, to capture chromatophore dynamics accurately. Sharp, in-focus frames are critical, particularly for moving subjects, where higher shutter speeds help minimize motion blur. For reliable segmentation, each chromatophore should cover at least 10 pixels across its fully expanded diameter. Higher spatial resolution, with chromatophores covering around 50 pixels in diameter, is recommended if sub-chromatophore dynamics are of interest. Recording conditions should minimize background noise, and the water column should be as clear as possible, free of particles or debris. The water surface should be kept as calm and planar as possible to avoid optical artifacts. If wide-angle lenses or other optics that may introduce distortion are used, lens correction algorithms should be applied during preprocessing to compensate for the optical distortions. For long-term tracking applications (e.g., developmental studies), frequent imaging sessions are recommended. Newly differentiated chromatophores are initially light colored (e.g., yellow) and thus visually distinct from mature chromatophores (which are dark); over days to weeks, however, the light chromatophores darken and become increasingly difficult to differentiate from older ones. Recording at appropriate and regular intervals thus helps track individual chromatophores across developmental stages and improves the reliability of long-term analyses. Following these recommendations will help segmentation, tracking, and analysis with CHROMAS.

## Methods

### Implementation details

CHROMAS is written for GNU/Linux, Windows, and MacOS operating systems in the Python programming language, requiring Python 3.9 or higher. The software relies on several key libraries for various functionalities. *NumPy* (**Harris et al., 2020**), *SciPy* (**Virtanen et al., 2020**), *pandas* (**McKinney, 2010**), and *xarray* (**Hoyer and Hamman, 2017**) are used for most of the numerical computations, utilizing labeled multi-dimensional arrays. *Dask* (**Rocklin, 2015**; **Dask Development Team, 2016**) is used for parallel computing and distributed task scheduling. Image processing and computer vision tasks are performed using *OpenCV* (**Bradski, 2000**), *scikit-image* (**van der Walt et al., 2014**), and *decord* (**Howard et al., 2019**) for efficient video decoding. *scikit-learn* (**Pedregosa et al., 2011**) is used for data preprocessing, classification models and clustering, *PyTorch* (**Paszke et al., 2019**) and *torchvision* (**Marcel and Rodriguez, 2010**) for deep-learning-based segmentation models, and *albumentations* (**Buslaev et al., 2020**) for image augmentation used in the training of these models. Visualization

is accomplished using *Matplotlib* (*Hunter, 2007*) for creating static, animated, and interactive visualizations, and *Bokeh* (*Bokeh Development Team, 2014*) for interactive dashboards visualizing computations. Data storage is performed using *Zarr* (*Miles et al., 2020*) for chunked, compressed, *N*-dimensional arrays. This allows for efficient storage and access of large, chunked, and compressed *N*-dimensional arrays. The command-line interface is created using *Click* (*Ronacher, 2014*), while *tqdm* (*tqdm Developers, 2016*) provides progress bar functionality and the TUI is built using *Trogon* (*Freeman, 2022*). Documentation is generated using *Sphinx* (*Sphinx Team, 2007*). *nbsphinx* (*Grünwald, 2017*) is used to include Jupyter notebooks in the documentation, with *jupyter-client* and *ipykernel* providing Jupyter notebook support (*Jupyter Development Team, 2015*). *Pandoc* (*MacFarlane, 2006*) is used for document conversion and *rpy2* (*Gautier, 2008*) is used for exporting data in R formats.

Selected sample data accompany the tutorials.

## Training machine learning models to segment chromatophores

### Data augmentation
To reduce the amount of manually annotated images needed for training segmentation models, data augmentation techniques are employed to expand the size of the training dataset. This approach was also used in the training of the enclosed models. The *albumentations* library (*Buslaev et al., 2020*) is used for this purpose, offering a variety of transformations including resizing, shifting, scaling, rotation, horizontal and vertical flips, RGB-value shifting, brightness and contrast adjustment, downscaling, random shadows, FancyPCA, and perspective transforms.

### Models
CHROMAS offers three distinct types of segmentation classifiers, each tailored to accommodate varying levels of visual variability in the dataset, processing speed, and available amounts of training data.

The first and simplest approach is a lookup table for color values in a specified color space. The training of this classifier involves selecting a color space (e.g. RGB, HSV, or Lab) and mapping a small set of representative color values to specific classes (e.g., background, chromatophore). Once the data points are mapped, all possible color values are classified in advance based on these inputs, generating a lookup table that assigns pixel color values to the respective classes. This method is particularly well-suited for videos where the chromatophores have distinctly different color values from the surrounding skin.

Some of our datasets exhibited significant visual variability, stemming from changes in illumination (LED intensity, blue light, or lighting angle), optical adjustments such as aperture, and the skin's varying light reflectance and transmittance. For example, in *E. berryi*, internal structures visible through the transparent skin contribute to making certain skin regions darker. To deal with these issues, we developed two other methods.

The second option uses a random forest classifier, a computationally efficient approach when lacking a Graphical Processing Unit (GPU) that requires relatively little training data and sometimes outperforms other methods when segmenting very small chromatophores (less than 10 pixels in size). However, its performance remains strongly tied to the lighting conditions of the training data.

The third approach uses deep learning models, specifically Fully Convolutional Networks (*Long et al., 2015*), DeepLabV3 (*Chen et al., 2017*), and U-Net (*Ronneberger et al., 2015*), with either ResNet50, ResNet101 (*He et al., 2016*), or MobileNetV3-Large (*Howard et al., 2019*) as backbones. This neural network-based method requires more training data, especially at the beginning. Our pre-trained models (trained on manually annotated images of *E. berryi* and *S. officinalis* using Cross Entropy loss), however, are designed to function right away, or to serve as robust baselines for fine-tuning, thus reducing the need for large datasets. While this approach requires access to an NVIDIA GPU, it generalizes well across conditions, making it the most flexible and powerful option.

## Registration details
The Lucas–Kanade optical flow algorithm, used for video registration, tracks a sparse set of points across the video. These points are initially defined in the first frame as centers of mass of small chromatophores. The process starts by identifying chromatophore regions through segmentation of the

first frame, with an offset calculated to exclude points near the edges based on a percentage of the image dimensions. Candidates are filtered based on shape properties such as eccentricity, solidity, and area and are subsampled evenly across a grid to ensure that tracking points are well distributed.

After selecting and tracking these points throughout the video, displacement maps of a finer equidistant grid are interpolated using the moving-least-squares algorithm. Full displacement maps are then calculated by linearly interpolating between these points, providing comprehensive registration of the video frames.

### Stitching reprojection error

The stitching algorithm is used to align each masterframe in a dataset to every other masterframe. The accuracy of these non-symmetric mappings is quantified by calculating the reprojection error, as developed in *Reiter et al., 2018*. Specifically, all points within a mask—either the animal's body if it is fully visible or a selected region of interest—of masterframe A are mapped to the reference frame of masterframe B using the A-to-B map, and then mapped back to the reference frame of A using the B-to-A map. The reprojection error is defined as the Euclidean distance between the original points and their remapped counterparts.

### Clustering motion correlation data

**PCA** is a statistical technique used to reduce data dimensionality while retaining as much variance as possible (*Jolliffe and Cadima, 2016*). It identifies principal components—orthogonal directions that capture the most variance in the data. These components represent independent patterns of variation within the dataset. In the context of chromatophore anisotropy, motor neurons influence subsets of radial muscles, creating distinct patterns of contraction and expansion. By applying PCA to the activity data of the slices (and indirectly, the radial muscles), we could uncover the primary directions of variance, corresponding to the independent influence of individual motor neurons (to the extent that they were recruited independently of one another at least some of the time). We used the **elbow method** to set the number of components worth retaining: in a cumulative-explained-variance versus number of components, the value of $x$ at the inflection point beyond which additional components captured less and less additional variance—was chosen. Essentially, each retained principal component reflects a distinct source of coordinated muscle activity, likely driven by a separate motor neuron.

After identifying the number of principal components using PCA, we apply **ICA** to further identify the underlying sources of activity within our dataset. ICA attempts to separate the data into statistically independent sources (*Bell and Sejnowski, 1995*; *Hyvärinen and Oja, 2000*). In the context of our study, each motor neuron can be thought of as an independent source of muscle activity, which ICA is designed to isolate, even if the signals overlap spatially or temporally. This two-step approach—using PCA to find the number of components and then ICA to extract the independent sources—allowed us to capture both the number and activity of motor neurons acting on a single chromatophore's radial muscles, ultimately providing a clearer picture of the likely neuromuscular control behind their dynamic shape modulation.

To identify motor units (a motor neuron and the chromatophores it innervates), we first perform ICA on each individual chromatophore to extract the independent components (ICs) which correspond to motor neuron signals influencing that chromatophore's activity. To group the chromatophores innervated by the same motor neuron, we use clustering algorithms such as **Affinity Propagation** (*Frey and Dueck, 2007*) or **HDBSCAN** (*Campello et al., 2013*) to the ICs across all chromatophores, clustering by covariation. AP or HDBSCAN is particularly well suited to this task because they determine automatically the optimal number of clusters based on the structure of the data, are capable of handling high-dimensional data, accommodate noise, and detect subtle relationships. This method allows us to identify putative common motor neurons and map their influence across multiple chromatophores, giving us an understanding of the overall motor units controlling the dynamic patterns of chromatophore activity.

## Acknowledgements

We thank F Bayer for assistance in building the experimental setup; F Kretschmer for optimizing the camera control, recording software, and Git workflows; F Vollrath and S Junek for help with image acquisition; P Musset for help with high-performance computing; L Jürgens, S Schwind, LE. Reyes

de Frey, D Burgard, M Landler, M Minde S, Kranz P, Dominiczak NK, Vogt G, Wexel S, Dizdarevic, and E Northrup for animal care; and T Woo, X Liang, D Evans, M Elmaleh, and other members of the Laurent laboratory for constructive exchanges. We thank Alice Perenzin and Antje Berken for grant management and scientific coordination. This research was funded by the Max Planck Society (GL), by the LOEWE Schwerpunkt CMMS (State of Hesse) (GL), and by the European Union (ERC grant CAMOUFLAGE, 10114150) (GL).

## Additional information

### Funding

| Funder | Grant reference number | Author |
|--------|------------------------|--------|
| Max-Planck-Gesellschaft | | Gilles Laurent |
| European Research Council | 10.3030/101141501 | Gilles Laurent |
| LOEWE Schwerpunkt CMMS | | Gilles Laurent |

The funders had no role in study design, data collection, and interpretation, or the decision to submit the work for publication. Open access funding provided by Max Planck Society.

### Author contributions

Johann Ukrow, Conceptualization, Software, Validation; Mathieu DM Renard, Conceptualization, Data curation, Validation, Visualization, Methodology, Writing – original draft; Mahyar Moghimi, Software; Gilles Laurent, Conceptualization, Resources, Supervision, Funding acquisition, Project administration, Writing – review and editing

### Author ORCIDs

Johann Ukrow ⓘ https://orcid.org/0009-0009-4933-7871
Mathieu DM Renard ⓘ https://orcid.org/0009-0000-0801-7438
Gilles Laurent ⓘ https://orcid.org/0000-0002-2296-114X

### Ethics

All research and animal care procedures were carried out in accordance with the institutional guidelines that are in compliance with national and international laws and policies (DIRECTIVE 2010/63/EU; German animal welfare act; FELASA guidelines). The study was approved by the appropriate animal welfare authority (Dr. Vet. Med. E. Simon. Regierungspräsidium Darmstadt, Germany) under approval number V54-19c20/15-F126/1025.

Reviewer #1 (Public review): https://doi.org/10.7554/eLife.106509.3.sa1
Reviewer #2 (Public review): https://doi.org/10.7554/eLife.106509.3.sa2
Author response https://doi.org/10.7554/eLife.106509.3.sa3

## Additional files

### Supplementary files

MDAR checklist

### Data availability

CHROMAS is distributed via the pypi package index (https://pypi.org/project/chromas/) and is publicly released on GitLab (https://doi.org/10.17617/1.pa38-mh49) under the 3-Clause BSD License. The documentation is hosted on GitLab. The data used to train the segmentation models, the trained models, and example videos for the tutorial can be found at https://public.brain.mpg.de/Laurent/Chromas2025/.

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

# Appendix 1

**Appendix 1—key resources table**

| Reagent type (species) or resource | Designation | Source or reference | Identifiers | Additional information |
|---|---|---|---|---|
| Strain, strain background | *Euprymna berryi* (bobtail squid) | Laurent Lab, Max Planck Institute for Brain Research | | Lab-reared; used in developmental tracking and anisotropy analysis |
| Strain, strain background | *Sepia officinalis* (European cuttlefish) | Laurent Lab, Max Planck Institute for Brain Research | | Used for high-density chromatophore datasets |
| Software, algorithm | CHROMAS | This paper; https://doi.org/10.17617/1.pa38-mh49 | | Pipeline for chromatophore segmentation and analysis |
| Software, algorithm | Python | Python Software Foundation; https://www.python.org/ | | Version 3.9+ |
| Software, algorithm | PyTorch | *Paszke et al., 2019*; https://pytorch.org/ | | For segmentation models |
| Software, algorithm | Torchvision | *Marcel and Rodriguez, 2010*; https://pytorch.org/vision/ | | Model architectures |
| Software, algorithm | OpenCV | *Bradski, 2000*; https://opencv.org/ | | Image and video processing |
| Software, algorithm | scikit-learn (1.6.1) | *Pedregosa et al., 2011*; https://scikit-learn.org/ | | Clustering and dimensionality reduction |
| Software, algorithm | scikit-image (0.24.0) | *van der Walt et al., 2014*; https://scikit-image.org/ | | Image processing |
| Software, algorithm | albumentations (1.4.24) | *Buslaev et al., 2020*; https://albumentations.ai/ | | Data augmentation |
| Software, algorithm | xarray (2024.11.0) | *Hoyer and Hamman, 2017*; http://xarray.pydata.org/ | | Labeled multi-dimensional arrays |
| Software, algorithm | Zarr (2.18.4) | *Miles et al., 2020*; https://zarr.readthedocs.io/ | | Chunked data storage |
| Software, algorithm | Dask (2024.12.1) | *Dask Development Team, 2016*; https://www.dask.org/ | | Parallel and distributed processing |
| Software, algorithm | decord | *Wang, 2019*; https://github.com/dmlc/decord | | Efficient video loading |
| Software, algorithm | Matplotlib (3.10.0) | *Hunter, 2007*; https://matplotlib.org/ | | Visualization |
| Software, algorithm | Bokeh (3.6.2) | *Bokeh Development Team, 2014*; https://bokeh.org/ | | Interactive dashboards |

*Appendix 1 Continued on next page*

*Appendix 1 Continued*

| Reagent type (species) or resource | Designation | Source or reference | Identifiers | Additional information |
|---|---|---|---|---|
| Software, algorithm | Click (8.1.8) | *Ronacher, 2014*; https://click.palletsprojects.com/ | | CLI for CHROMAS |
| Software, algorithm | tqdm (4.67.1) | *tqdm Developers, 2016*; https://github.com/tqdm/tqdm | | Progress bars |
| Software, algorithm | Trogon (0.3.0) | *Freeman, 2022*; https://github.com/Textualize/trogon | | Terminal GUI |
| Software, algorithm | Sphinx | *Sphinx Team, 2007*; https://www.sphinx-doc.org/ | | Documentation generation |
| Software, algorithm | nbsphinx (0.9.6) | *Grünwald, 2017*; https://github.com/spatialaudio/nbsphinx | | Integrates Jupyter notebooks in Sphinx |
| Software, algorithm | jupyter-client | *Jupyter Development Team, 2015*; https://jupyter.org/ | | Jupyter support |
| Software, algorithm | Pandoc (2.4) | *MacFarlane, 2006*; https://pandoc.org/ | | Document conversion |
| Software, algorithm | rpy2 | *Gautier, 2008*; https://github.com/rpy2/rpy | | R–Python bridge |
| Other | Pre-trained chromatophore segmentation models | This paper; https://public.brain.mpg.de/Laurent/Chromas2025/ | | Trained on *E. berryi* and *S. officinalis* |
| Other | Workstation (Intel i9-10900X+RTX A4000) | Laurent Lab, Max Planck Institute for Brain Research | | Used for performance benchmarking |
| Other | Laptop (Intel i5-1135G7, no GPU) | Laurent Lab, Max Planck Institute for Brain Research | | Used to demonstrate pipeline performance on CPU |

