## [Editor Report · eLife Assessment]

The open-source software Chromas tracks and analyses cephalopod chromatophore dynamics. The software features a user-friendly interface alongside detailed instructions for its application, with **compelling** exemplary applications to two widely divergent cephalopod species, a squid and a cuttlefish, over time periods large enough to encompass new chromatophore development among existing ones. It demonstrates accurate tracking of the position and identity of each chromatophore. The software and methods outlined therein will become an **important** tool for scientists tracking dynamic signaling in animals.

---

## [Referee Report · Reviewer #1 (Public review)]

Summary:

This study provides comprehensive instructions for using the chromatophore tracking software, Chromas, to track and analyse the dynamics of large numbers of cephalopod chromatophores across various spatiotemporal scales. This software addresses a long-standing challenge faced by many researchers who study these soft-bodied creatures, known for their remarkable ability to change colour rapidly. The updated software features a user-friendly interface that can be applied to a wide range of applications, making it an essential tool for biologists focused on animal dynamic signalling. It will also be of interest to professionals in the fields of computer vision and image analysis.

Strengths:

This work provides detailed instructions for this tool kit along with examples for potential users to try. The Gitlab inventory hosts the software package, installation documentation, and tutorials, further helping potential users with a less steep learning curve.

Weaknesses:

The evidence supporting the authors' claims is solid, particularly demonstrated through the use of cuttlefish and squid. However, it may not be applicable to all coleoid cephalopods yet, such as octopuses, which have an incredibly versatile ability to change their body forms.

Comments on revisions:

I am pleased to see the more detailed version of this useful tool along with tutorials designed for diverse users who are interested in animal dynamic colouration. This study provides detailed instructions for using the chromatophore tracking software Chromas to track and analyse the dynamics of large numbers of cephalopod chromatophores across various spatiotemporal scales. The software features a user-friendly interface that is highly compelling and can be applied to a wide range of applications.

---

## [Referee Report · Reviewer #2 (Public review)]

Summary:

The authors developed a computational pipeline named CHROMAS to track and analyze chromatophore dynamics, which provides a wide range of biological analysis tools without requiring the user to write code.

Strengths:

(1) CHROMAS is an integrated toolbox that provides tools for different biological tasks such as: segment, classify, track and measure individual chromatophores, cluster small groups of chromatophores, analyze full-body patterns, etc.

(2) It could be used to investigate different species. The authors have already applied it to analyze the skin of the bobtail squid Euprymna berryi and the European cuttlefish Sepia officinalis.

(3) The tool is open-source and easy to install. The paper describes in detail the experiment requirements, command to complete each task and provides relevant sample figures, which are easy to follow.

Weaknesses:

(1) There are some known limitations for the current version. The users should read the "Discussion" chapter carefully before preparing their experiments and using CHROMAS.

---

## [Author Response]

The following is the authors’ response to the original reviews.

**Reviewer #1 (Public review):**
Summary:This study provides comprehensive instructions for using the chromatophore tracking software, Chromas, to track and analyse the dynamics of large numbers of cephalopod chromatophores across various spatiotemporal scales. This software addresses a long-standing challenge faced by many researchers who study these soft-bodied creatures, known for their remarkable ability to change colour rapidly. The updated software features a user-friendly interface that can be applied to a wide range of applications, making it an essential tool for biologists focused on animal dynamic signalling. It will also be of interest to professionals in the fields of computer vision and image analysis.Strengths:This work provides detailed instructions for this toolkit along with examples for potential users to try. The Gitlab inventory hosts the software package, installation documentation, and tutorials, further helping potential users with a less steep learning curve.Weaknesses:The evidence supporting the authors' claims is solid, particularly demonstrated through the use of cuttlefish and squid. However, it may not be applicable to all coleoid cephalopods yet, such as octopuses, which have an incredibly versatile ability to change their body forms.

The reviewer is right to highlight this limitation. We clarified, in the revised manuscript, that CHROMAS relies on the assumption that chromatophore activity occurs primarily in a plane — a condition that is valid most of the time in squid and cuttlefish, where the majority of skin deformations are in-plane (with small occasional papillae). In cephalopods such as octopuses, however, in which the skin may undergo large 3-dimensional deformations through the action of papillary musculature, this assumption may not always hold. Although octopods’ bodies are more spherical (less flat) than those of squid and cuttlefish, CHROMAS should still be usable and useful if applied to smaller skin areas, especially because chromatophore density is often even higher in octopoda than in sepiidae.

We added the following paragraph in the discussion:

Another known limitation concerns the biological assumptions underlying the current version of CHROMAS. The pipeline is designed for surfaces that remain reasonably planar and undergo deformations primarily in two dimensions. In cephalopods such as octopuses, in which the skin can undergo substantial three-dimensional morphological changes, analysing chromatophore dynamics may require complementary three-dimensional tracking of the skin surface to correct for out-of-plane deformations and maintain accurate measurement of chromatophore activity.

**Reviewer #2 (Public review):**
Summary:The authors developed a computational pipeline named CHROMAS to track and analyse chromatophore dynamics, which provides a wide range of biological analysis tools without requiring the user to write code.Strengths:(1) CHROMAS is an integrated toolbox that provides tools for different biological tasks such as: segment, classify, track and measure individual chromatophores, cluster small groups of chromatophores, analyse full-body patterns, etc.(2) It could be used to investigate different species. The authors have already applied it to analyse the skin of the bobtail squid Euprymna berryi and the European cuttlefish Sepia officinalis.(3) The tool is open-source and easy to install. The paper describes in detail the command format to complete each task and provides relevant sample figures.Weaknesses:(1) The generality and robustness of the proposed pipeline need to be verified through more experimental evaluations. For example, the implementation algorithm depends on relatively specific or obvious image features, clean backgrounds, and objects that do not move too fast.(2) The pipeline lacks some kind of self-correction mechanism. If at one moment there is a conflicting match with the previous frames, how does the system automatically handle it to ensure that the tracking results are accurate over a long period of time?

We thank the reviewer for raising this important point. CHROMAS does rely on relatively clean imaging conditions for optimal performance. However, the computational features of the pipeline — segmentation, tracking, and downstream analysis — have been designed to perform reliably as long as the segmentation models are trained on frames that reflect the diversity of the dataset (e.g., variations in lighting or minor background noise). It is correct, however, that acquiring the necessary quality of input data is both important and non-trivial. The pipeline is designed to work best with high-resolution footage of chromatophores under clear imaging conditions — specifically, with minimal water surface distortion, minimal particulate matter in the water column, and stable focus.

To mitigate issues arising from motion blur or focus loss, CHROMAS includes an automatic frame quality control step that detects and discards frames that are out of focus, including those where the animal moves too fast for reliable tracking.

To assist future users, we have now added a section under Discussion detailing the recommended recording conditions and video characteristics for effective analysis with CHROMAS. It reads:

Recommended Video Parameters for Optimal Use of CHROMAS

The performance of CHROMAS depends on the quality of the input videos. Although the pipeline analyses each frame independently and has no frame rate requirement, we recommend recording at 20 frames per second at least, to capture chromatophore dynamics accurately. Sharp, in-focus frames are critical, particularly for moving subjects, where higher shutter speeds help minimize motion blur. For reliable segmentation, each chromatophore should cover at least 10 pixels across its fully expanded diameter. Higher spatial resolution, with chromatophores covering around 50 pixels in diameter, are recommended if sub-chromatophore dynamics are of interest. Recording conditions should minimize background noise, and the water column should be as clear as possible, free of particles or debris. The water surface should be kept as calm and planar as possible to avoid optical artifacts. If wide-angle lenses or other optics that may introduce distortion are used, lens correction algorithms should be applied during preprocessing to compensate for the optical distortions. For long-term tracking applications (e.g., developmental studies), frequent imaging sessions are recommended. Newly differentiated chromatophores are initially light colored (e.g., yellow) and thus visually distinct from mature chromatophores (which are dark); over days to weeks, however, the light chromatophores darken and become increasingly difficult to differentiate from older ones. Recording at appropriate and regular intervals thus helps track individual chromatophores across developmental stages and improves the reliability of long-term analyses. Following these recommendations will help segmentation, tracking, and analysis with CHROMAS.

CHROMAS does not implement an active self-correction mechanism in the sense of real-time error recovery. Yet, several steps are in place to ensure the reliability of registration and tracking over time. During registration, a set of points is tracked across frames using optical flow. If the displacement of a point between two frames exceeds a biologically plausible threshold, that point is automatically discarded from the registration calculation to prevent error propagation. If too many points are discarded, the registration step fails, preventing the acceptance of a poor alignment.

In addition, masterframes (the averages of all aligned frames in a chunk) are generated at the end of the registration process to enable the visual verification of the quality of the mapping.

During stitching, CHROMAS calculates reprojection errors between chunks, providing a quantitative measure of stitching validity and allowing users to detect and correct potential mismatches.

We have revised the Results section to explicitly highlight the error-checking mechanisms implemented during registration and stitching to maintain tracking accuracy over time.

**Reviewer #1 (Recommendations for the authors):**
(1) Figures 2, 3, 5, 6, 8 showed the bobtail squid, however, all command lines for these figures were referred to "sepia_example.dataset".

We thank the reviewer for noticing this inconsistency. We have corrected the labeling of the dataset name in the command line examples from "sepia_example.dataset" to the neutral term "example.dataset" to avoid any confusion regarding the species used in the figures.

(2) It's excellent that Chromas includes a manual pre-alignment function. However, it's unclear how the authors determined the registration of selected chromatophores across different ages in the long-term tracking session. Given the rapid growth of cephalopods and presumably skin expansion with increased chromatophores, it would be helpful to provide more details or examples on this process.

The manual pre-alignment function provides an interactive interface allowing the user to select a set of matching chromatophores across frames from different developmental stages. The accuracy of this process depends on the user's ability to recognize individual chromatophores reliably over time. Critically, it is not necessary to identify all those chromatophores; a representative subset is sufficient to interpolate the spatial mapping and align the surrounding chromatophores.

To limit the potential challenges associated with chromatophore development, frequent imaging sessions (every few days) are recommended initially. Excessive intervals between recordings can result in relative displacements among existing chromatophores and the sudden appearance of newly matured chromatophores, both of which complicate manual matching.

It should be noted that these challenges are not limitations of the CHROMAS pipeline itself, but rather relate to experimental design choices that affect the quality and traceability of the dataset. The exact parameters (e.g., size/duration of the datasets, spatial resolution, frame rate and intervals between recording sessions) to be used must be adapted to each experimental animal, each age, and ultimately, each question.

Recommended video acquisition parameters, including guidance on recording frequency for long-term chromatophore tracking, have been added to the Discussion section.

**Reviewer #2 (Recommendations for the authors):**
(1) More detailed information should be given, such as operating system requirements, camera frame rate requirements, target size and speed limitations, when chunking videos into usable segments, the minimum length of each segment, etc.

CHROMAS is platform-independent and requires only a functioning Python 3.9+ environment, regardless of the operating system or OS version, as described in “Methods – Implementation details”.

Although CHROMAS does not require specific frame rates and because it analyses each frame independently, the quality of each image—and thus of imaging parameters—is critical to enable reliable chromatophore segmentation. If an animal remains relatively calm during recording, low shutter speeds will be adequate for image sharpness. Conversely, if the animal moves frequently or rapidly, it will be preferable to use a higher frame rate and a higher shutter speed to minimize motion blur. Recording parameters should therefore be adjusted accordingly, primarily to optimize image clarity and maintain frames in sharp focus.

The frame rate should be sufficiently high also to capture the fast dynamics of chromatophore expansions and contractions. Although the pipeline has no specific frame rate requirement, we recommend image rates of at least 20 frames per second to sample the temporal patterns of chromatophore activity adequately, based on biological considerations.

Each chromatophore should be represented by a sufficiently large number of pixels in each recorded image to enable the reliable estimation of its size, shape, and dynamics. If the spatial resolution is too low, individual chromatophores may appear as small pixel clusters, reducing the accuracy of area and shape measurements and introducing quantization artifacts. Based on our experience, we recommend recording conditions that result in each chromatophore covering at least 10 pixels across its diameter when fully expanded to ensure accurate segmentation and quantitative whole-chromatophore analysis. For sub-chromatophore motion analysis, we recommend a minimum of 50 pixels across the fully expanded diameter.

These considerations relate to optimizing biological sampling and image quality for analysis, and are not technical requirements imposed by CHROMAS itself.

We added a Discussion section outlining the recommended recording conditions and video parameters to facilitate effective use of CHROMAS.

(2) This pipeline does not include functionality to correct for lens distortion, which may affect the results when accurate measurement of single chromatophore morphology is required.

We thank the reviewer for this observation. We agree that lens distortion can affect the accurate measurement of chromatophore morphology if present. However, the current datasets analysed with CHROMAS were recorded using a long macro lens with minimal distortion, and visual inspections as well as quantitative assessments of chromatophore geometry did not indicate measurable optical deformation. We acknowledge that for other imaging setups —particularly those relying on the use of wide-angle lenses— lens distortion could introduce artifacts. In such cases, we recommend applying standard lens distortion correction during preprocessing, prior to analysis with CHROMAS.

We have also addressed this point in the newly added section under the Discussion.

(3) How to perform expansion for single chromatophores shown in Figure 6, and how to keep the expansion area consistent?

The graph in Figure 6 illustrates the expansion of a single chromatophore over time and was generated entirely using the "areas" command and visualization tools available within CHROMAS.

Spatial consistency is maintained because CHROMAS, through its registration and area extraction steps, tracks the identity of each chromatophore across the video, allowing the same individual to be followed reliably over time.

(4) Tables 1 and 2: it's better to add the units of the values in each column.

We thank the reviewer for the suggestion. We have added the appropriate units to each column in Tables 1 and 2 to improve clarity.